# Disubstituted Meldrum’s Acid: Another Scaffold with SuFEx-like Reactivity

**DOI:** 10.3390/molecules30173534

**Published:** 2025-08-29

**Authors:** Baoqi Chen, Zhenguo Wang, Xiaole Peng, Jijun Xie, Zhixiu Sun, Le Li

**Affiliations:** PCFM Lab and GDHPRC Lab, School of Chemistry, Sun Yat-sen University, Guangzhou 510275, China; benchan93@163.com (B.C.); wangzhg28@mail2.sysu.edu.cn (Z.W.); pengxle@mail2.sysu.edu.cn (X.P.); xiejj35@mail2.sysu.edu.cn (J.X.); sunzx1996@163.com (Z.S.)

**Keywords:** disubstituted Meldrum’s acid, esterification, amidation, decarboxylation, SuFEx

## Abstract

Sulfur Fluoride Exchange (SuFEx) chemistry represents an emerging class of click reactions that has found broad applications in drug discovery and materials science. Traditionally, SuFEx reactivity has been regarded as the exclusive privilege of sulfur and fluorine. Accordingly, the scaffolds exhibiting SuFEx-like reactivity without sulfur or fluorine have remained underdeveloped. Indeed, SuFEx reactions may represent a more generalizable mode of chemical reactivity. By enhancing the electrophilicity of the carbonyl group and increasing the steric hindrance around the carbon center, we identified disubstituted Meldrum’s acid as a novel carbon-based scaffold with SuFEx-like reactivity. Various O-, S-, and N-nucleophiles are viable exchange partners in the presence of Barton’s base or DBU. In addition to the original method, a catalytic protocol was developed and successfully applied to drug derivatization, including the gram-scale modification of acetaminophen.

## 1. Introduction

The concept of Sulfur Fluoride Exchange (SuFEx) [1,2,3,4,5,6], first introduced by Sharpless and coworkers, represents a unique class of nucleophilic exchange reactions in which stable sulfur(VI) fluoride compounds can be selectively activated under specific environments [7,8,9,10,11,12], in the presence of promoters [13,14], or under catalytic conditions [15,16,17,18,19,20] to form covalent linkages. This "dormant awakening reactivity" has enabled broad applications across chemical biology [21,22,23,24,25,26,27], drug discovery [28,29,30,31,32,33,34,35,36,37,38], and materials science [39,40,41,42,43,44,45,46,47,48,49,50,51,52]. Initially, SuFEx reactivity was attributed to the distinctive characteristics of the S(VI)–F bond (Figure 1a). Following this paradigm, a variety of building blocks containing an S–F bond have been developed, greatly expanding the SuFEx toolbox [53,54,55,56,57,58,59,60,61,62,63,64,65]. However, the reliance on either sulfur or fluorine elements in SuFEx chemistry has been challenged by recent discoveries. For instance, Zuilhof and coworkers introduced the concept of sulfur-phenolate exchange (SuPhenEx) in their latest work, demonstrating that S(VI) centers can undergo efficient exchange reactions without the involvement of fluorine (Figure 1b) [66,67,68,69]. In 2023, Moses and coworkers discovered a set of phosphorus-based exchange reactions, termed Phosphorus Fluoride Exchange (PFEx), wherein P(V) fluorides exhibit similar latent reactivity with nucleophiles (Figure 1c) [70,71,72,73]. These findings suggest that SuFEx-like reactivity may not be limited to S- or F-based systems, but rather represent a more general reactivity mode for robust covalent bond formation. Therefore, it is highly desirable to discover new scaffolds capable of mimicking SuFEx-type reactivity. Herein, we report a carbon-based scaffold, disubstituted Meldrum’s acid, that exhibits SuFEx-like reactivity under mild conditions (Figure 1d).

Acyl fluorides were the original candidates for this project. However, due to the relatively low steric hindrance around the acyl carbon, acyl fluorides, despite their high exchange reactivity, exhibited significantly lower hydrolytic stability compared to S(VI)–fluorides. This limitation led us to shift our focus to other carbonyl compounds with high intrinsic reactivity. Ultimately, disubstituted Meldrum’s acid was identified as a promising candidate. Although Meldrum’s acid may seem unrelated to SuFEx-like reactivity at first glance, its cyclic ester structure [74,75,76,77,78] endows it with both high electrophilicity and hydrolytic stability. This pairing of electrophilicity and stability strongly resonates with the design principles of conventional SuFEx reagents. (Figure 1). In addition, disubstituted Meldrum’s acids are highly tunable building blocks and can be readily prepared from Meldrum’s acid via alkylations, Michael additions, and reductive Knoevenagel condensations [79,80]. This ease of double substitution imparts remarkable structural diversity to disubstituted Meldrum’s acids, which in turn facilitates multidimensional connectivity via nucleophilic ring-opening reactions. Meanwhile, disubstituted Meldrum’s acids possess markedly enhanced stability relative to their less substituted counterparts. However, several challenges persist in the proposed decarboxylative nucleophilic ring-opening reaction. Notably, while the nucleophilic ring-opening of disubstituted Meldrum’s acids occurs readily [79,81,82,83,84,85,86,87,88,89], only a very limited number of them undergo decarboxylation under mild conditions [90,91,92]. In many cases, decarboxylation still requires elevated temperatures or the use of excess reagents to achieve efficient conversion [93,94,95,96,97,98,99,100,101]. These existing conditions fail to meet the standards of SuFEx chemistry. Therefore, a facile protocol compatible with more complex synthetic settings is highly desirable.

## 2. Results

Initially, we chose **2a** as a nucleophile to investigate the nucleophilic exchange reactions of **1a**. A control experiment conducted in the absence of any base showed no formation of the expected product, and **1a** remained intact (Table 1, entry 1). In addition, either inorganic bases or weaker organic bases resulted in negligible formation of **3a** (Table 1, entries 2–8). Further investigation revealed that **1a** was completely consumed when a strong inorganic base, such as cesium carbonate, potassium carbonate, or potassium phosphate, was used. In these cases, the nucleophilic ring-opening product, malonate half ester **3a’** was obtained instead of **3a**. In the other cases using sodium carbonate, 4-dimethylaminopyridine (DMAP), triethylamine (TEA), and *N,N*-diisopropylethylamine (DIPEA), most of 1a remained unchanged. Gratifyingly, both nucleophilic substitution and decarboxylation proceeded smoothly at room temperature in the presence of 1,8-diazabicyclo[5.4.0]undec-7-ene (DBU) or Barton’s base (2-tert-Butyl-1,1,3,3-tetramethylguanidine, BTMG) (Table 1, entry 9–10). These results were somewhat unexpected, as the decarboxylation of malonate half-esters typically requires elevated temperatures. Next, we extended the reaction time from 4 h to 12 h and evaluated the compatibility of this reaction with a variety of common solvents (Table 1, entries 11–16). To our delight, nearly quantitative yields were obtained in acetonitrile, toluene, tetrahydrofuran (THF), *N*-methyl-pyrrolidinone (NMP), and dimethyl sulfoxide (DMSO), while a slightly lower yield (87%) was observed in dichloromethane. To evaluate the potential applicability of this transformation in biological settings, aqueous DMSO solutions were tested, and satisfactory yields were obtained (Table 1, entries 17–18). The high efficiency of this transformation across various solvents aligned well with the principles of click chemistry.

With the optimal protocol in hand, the substrate scope was acid and phenolic nucleophiles were well tolerated and α,α-disubstituted phenyl acetates were obtained in excellent isolated yields (**3a**–**3y**) (Figure 2). For the Meldrum’s acid component, we focus on evaluating the substrates with a larger alkyl group. The results indicated that a variety of benzyl-substituted substrates afforded excellent yields (**3f**–**3j**). Notably, more sterically hindered alkyl substrates (**1b**–**1e**) also provided satisfactory results. For phenolic nucleophiles, we examined the influence of functional group tolerance, electronic properties, and steric hindrance. The method demonstrated excellent tolerance toward diverse functional groups. A wide array of *α,α*-disubstituted phenylacetates bearing diverse functional groups such as halogens (**3t**, **3x**, and **3y**), trifluoromethyl (**3l**), amino (**3n**), alkoxy (**3o** and **3w**), nitro (**3k**), ester (**3p**), aldehyde (**3q**), alkenyl **(3w**), and various heterocycles (**3m**, **3x**, and **3y**), were obtained in excellent yields. Although the method was generally insensitive to electronic effects, the most electron-deficient phenol, 4-nitrophenol (**2k**), and the most electron-rich one, 4-dimethylaminophenol (**2n**), required a more polar solvent, NMP, and a more concentrated condition (0.8 M) to complete the reaction at room temperature. In terms of steric effects, *ortho*-substituted substrates (**2s**, **2t**, and **2u**) and 1-naphthol (**2r**) reacted smoothly under standard conditions. The only exception was the highly sterically hindered 2,6-dimethylphenol (**2v**), which required heating at 60 °C to achieve full conversion.

Next, we turned our attention to the use of alcohols, thiols, and amines as nucleophilic partners. Several representative nucleophiles were selected for evaluation. Not surprisingly, no desired *α,α*-disubstituted acetate products were obtained when alcohols were used as nucleophiles under the standard conditions developed for phenols, even at 40 °C. However, small amounts of the desired products could be detected upon raising the reaction temperature to 60 °C in acetonitrile. Analysis of the reaction mixtures revealed that the major products under these conditions were the non-decarboxylated malonate half-esters. To improve the yield of the expected ester product, we employed NMP as the solvent to promote decarboxylation. Gratifyingly, this subtle adjustment successfully led to the formation of the desired esters (**4a**, **4b**, **4c**, and **4d**) in high yields from alcohol nucleophiles (Figure 3). Indeed, primary alcohols such as (4-bromophenyl)methanol and 3-phenylpropan-1-ol, as well as secondary diphenylmethanol, reacted smoothly at 60 °C. Furthermore, cyclohexanol, due to its increased steric hindrance, required the use of a less hindered base (DBU) in place of BTMG, along with a slightly higher temperature (80 °C) to afford the product **4d**. Interestingly, the reactivity of thiol and thiophenol more closely resembles that of phenols rather than alcohols. Thioesters (**4e** and **4f**) were delivered in excellent yields at room temperature. Encouraged by these results, we further increased the reaction temperature to 100 °C for amidations and confirmed that primary amines were viable nucleophiles under elevated temperature (Figure 3). The corresponding amide products (**4g**, **4h**, **4i**, and **4j**) were obtained at 100 °C in satisfactory yields. We also investigated a secondary amine (morpholine) as the nucleophile. As a result, **4k** was obtained in 62% isolated yield using the conditions for primary amines. As anticipated, the sterically hindered tert-butanol and tert-butylamine did not undergo nucleophilic exchange reactions and failed to yield the desired products **4l** and **4m** even at 100 °C.

After evaluating the scope of nucleophilic exchange partners, we sought to improve the practicality of the current methodology in scalable synthesis. To avoid the use of stoichiometric amounts of BTMG, we aimed to develop a catalytic protocol for this transformation. Acetaminophen, a widely used over-the-counter (OTC) fever and pain reliever, was selected as the model substrate. Gratifyingly, using a catalytic amount of BTMG in NMP at 60 °C for 12 h, the desired acylated product was obtained in 96% yield on a 2.12 g scale (Figure 2a). In addition, we are able to isolate the malonate half-ester **5** by using a modified protocol (Figure 2b). Subsequently, the Curtius rearrangement of **5** successfully delivered the valuable α-amino acid derivatives **6** in 76% yield (Figure 2c). Further investigation revealed that the isolation of the malonate half-ester **5** was not required. Accordingly, a one-pot synthesis of malonyl monoester amide **7** from **1a** and **2** was successfully achieved (Figure 2d).

Based on the experimental data we obtained, a plausible catalytic cycle is proposed. The reaction proceeds via decarboxylation of the malonate half-ester followed by reprotonation (Figure 3) [102,103]. The superior results obtained with BTMG, in contrast to inorganic bases, are likely due to its excellent solubility in various organic solvents, which is particularly important for facilitating the decarboxylation. This pathway is distinct from the ketene mechanism observed under thermal conditions (>150 °C) [104,105,106,107]. To further support this mechanistic proposal, the malonate half-ester **3a’** was treated with a stoichiometric amount of BTMG at room temperature. The decarboxylation of **3a’** occurred readily, affording the product **3a** (see Appendix A).

## 3. Materials and Methods

### 3.1. Reagents and General Methods

All reagents were purchased from commercial sources(Energy Chemical, Shanghai, China) and used without further purification unless otherwise noted. Acetonitrile, *N*-methyl-pyrrolidinone and dimethyl sulfoxide used in the reactions were anhydrous solvents purchased from commercial suppliers (Energy Chemical, Shanghai, China) and used without further drying (≥99.9%, LC-MS, Energy Chemical, China). The petroleum ether (Energy Chemical, Shanghai, China) used was in a boiling range of 60–90 °C. Other solvents were purified according to standard procedures [108]. All reactions were carried out with oven-dried glassware and monitored by thin-layer chromatography (0.20 mm HP-TLC silica gel 60 GF-254 plates, Leyan, Shanghai, China). Visualization was accomplished with UV light, and/or potassium permanganate, or 2% ninhydrin in ethanol stain followed by heating. Flash column chromatography was performed on 200–300 mesh silica gel (Leyan, Shanghai, China). Meldrum’s acid (2,2-dimethyl-1,3-dioxane-4,6-dione) and 5-methyl Meldrum’s acid (2,2,5-trimethyl-1,3-dioxane-4,6-dione) were purchased from Bide Pharm, Shanghai, China, and used without further purification. The deuterated chloroform (CDCl_3_) (Energy Chemical, Shanghai, China) used contains 0.03% (*v*/*v*) of tetramethylsilane (TMS).

^1^H, ^19^F, and ^13^C NMR spectra were recorded on a Bruker AVANCE III 400 MHz spectrometer (Bruker, Billerica, MA, USA) at 298 K and referenced to residual protium in the NMR solvent (CDCl_3_, δ 7.26, DMSO-*d*_6_, 2.50 in ^1^H NMR) and the carbon resonances of the solvent (CDCl_3_, δ 77.16, DMSO-*d*_6_, 39.52 in ^13^C NMR). Chemical shifts were reported in parts per million (ppm, δ) downfield from tetramethylsilane. NMR peaks are described as singlet (s), doublet (d), triplet (t), quartet (q), multiplet (m), heptet (hept), complex (comp), and approximate (app). High-resolution mass spectra (HRMS) were recorded on a Thermo Fisher Scientific’s Q Exactive UHMR Hybrid Quadrupole-Orbitrap Mass Spectrometer LC/MS (ESI/APCI) (Thermo Fisher, Waltham, MA, USA).

TEAF = triethylammonium formate, TEA = triethylamine, DIPEA = *N,N*-diisopropylethylamine, DBU = 1,8-diazabicyclo[5.4.0]undec-7-ene, DMAP = 4-dimethylaminopyridine, BTMG = 2-*tert*-butyl-1,1,3,3-tetramethylguanidine, NMI = *N*-methylimidazole, DPPA = diphenyl azidophosphate, HOBt = 1-hydroxybenzotriazole, EDC = 1-ethyl-3-(3-dimethylaminopropyl)carbodiimide, EA = ethyl acetate, DMF = *N,N*-dimethylformamide, DCM = dichloromethane, DCE = dichloroethane, DMSO = dimethyl sulfoxide, THF = tetrahydrofuran, NMP = *N*-methyl-pyrrolidinone, TLC = thin-layer chromatography, *v*/*v* = volume per volume, equiv = equivalent, w/o = without, rt = room temperature.

### 3.2. Synthetic Procedures

#### 3.2.1. A General Procedure for the 5-Substituted-5-methyl-1,3-dioxane-4,6-dione

General Procedure: Substituted-Meldrum’s acids were prepared according to the literature procedure [109]. 5-methyl Meldrum’s acid (1.00 equiv) and K_2_CO_3_ (1.30 equiv) were dissolved in anhydrous DMF, followed by the addition of alkyl bromide (1.20 equiv). The mixture was then heated to 60 °C for 12 h. After reaction completion, the mixture was diluted with 50 mL of EA and 50 mL of water. The organic phase was separated. The aqueous phase was further extracted with 2 × 50 mL of EA. The combined organic layer was separated, washed with saturated aqueous NaHCO_3_ solution and saturated aqueous NaCl solution, dried over anhydrous sodium sulfate, and concentrated in vacuo to afford the crude product. The crude product was purified by flash column chromatography to afford the purified product.

#### 3.2.2. A General Procedure for the Preparation of the Esters **3**

In a 4.0-mL vial, 5,5-substituted-Meldrum’s acid **1** (0.50 mmol, 1.00 equiv), BTMG (94.2 mg, 0.55 mmol, 1.10 equiv) were dissolved in anhydrous MeCN (1.25 mL). The phenol (0.55 mmol, 1.10 equiv) was added and stirred at rt for 12 h. After the reaction was completed, the mixture was diluted with EA (20 mL), washed with 2 M aqueous HCl solution (15 mL). The organic layer was separated, washed with saturated brine (20 mL), dried over anhydrous sodium sulfate, filtered, and concentrated in vacuo. The crude product was then purified by silica gel column chromatography.

*4-Bromophenyl isobutyrate* (**3a**): Compound **3a** was prepared according to the general procedure using **1a** (86.0 mg, 0.50 mmol, 1.00 equiv) and *p*-bromophenol (95.2 mg, 0.55 mmol, 1.10 equiv) at rt for 12 h. Purification by flash column chromatography (petroleum ether/ethyl acetate = 20/1, *v*/*v*) afforded 4-bromophenyl isobutyrate (**3a**) as a colorless oil (111.8 mg, 92%). R_f_ = 0.40 (petroleum ether/ethyl acetate = 40/1, *v*/*v*). **^1^H NMR** (400 MHz, CDCl_3_) δ 7.48 (d, *J* = 7.5 Hz, 2H), 6.97 (d, *J* = 7.5 Hz, 2H), 2.79 (hept, *J* = 6.9 Hz, 1H), 1.31 (d, *J* = 6.9 Hz, 6H). **^13^C NMR** (101 MHz, CDCl_3_) δ 175.3, 149.9, 132.4 (2C), 123.4 (2C), 118.7, 34.2, 18.9 (2C). **HRMS−ESI** (*m*/*z*) for C_10_H_11_BrO_2_ [M + Na]^+^: calcd 264.9834 (^79^Br), 266.9814 (^81^Br), found 264.9835 (^79^Br), 266.9814 (^81^Br).

*4-Bromophenyl 2,3-dimethylbutanoate* (**3b**): Compound **3b** was prepared according to the general procedure using **1b** (100.0 mg, 0.50 mmol, 1.00 equiv) and *p*-bromophenol (95.2 mg, 0.55 mmol, 1.10 equiv) at rt for 12 h. Purification by flash column chromatography (petroleum ether/ethyl acetate = 20/1, *v*/*v*) afforded 4-bromophenyl 2,3-dimethylbutanoate (**3b**) as a colorless oil (115.2 mg, 85%). **R*_f_*** = 0.40 (petroleum ether/ethyl acetate = 40/1, *v*/*v*). **^1^H NMR** (400 MHz, CDCl_3_) δ 7.5 (d, *J* = 8.8 Hz, 2H), 7.0 (d, *J* = 8.9 Hz, 2H), 2.5 (m, 1H), 2.1 (m, 1H), 1.3 (d, *J* = 7.1 Hz, 3H), 1.04 (d, *J* = 6.8 Hz, 1H), 1.00 (d, *J* = 6.7 Hz, 1H). **^13^C NMR** (101 MHz, CDCl_3_) δ 174.5, 149.9, 132.4 (2C), 123.4 (2C), 118.7, 46.2, 31.2, 20.7, 19.2, 13.7. **HRMS−ESI** (*m*/*z*) for C_12_H_15_BrO_2_ [M + Na]^+^: calcd 293.0147 (^79^Br), 295.0127 (^81^Br), found 293.0149 (^79^Br), 295.0127 (^81^Br).

*4-Bromophenyl 2,4-dimethylpentanoate* (**3c**): Compound **3c** was prepared according to the general procedure using **1c** (107.0 mg, 0.50 mmol, 1.00 equiv) and *p*-bromophenol (95.2 mg, 0.55 mmol, 1.10 equiv) at rt for 12 h. Purification by flash column chromatography (petroleum ether/ethyl acetate = 20/1, *v*/*v*) afforded 4-bromophenyl 2,4-dimethylpentanoate (**3c**) as a colorless oil (126.4 mg, 90%). **R*_f_*** = 0.40 (petroleum ether/ethyl acetate = 40/1, *v*/*v*). **^1^H NMR** (400 MHz, CDCl_3_) δ 7.48 (d, *J* = 8.8 Hz, 2H), 6.96 (d, *J* = 8.7 Hz, 2H), 2.84–2.64 (m, 1H), 1.83–1.64 (comp, 2H), 1.42–1.31 (m, 1H), 1.28 (d, *J* = 6.9 Hz, 3H), 0.97 (d, *J* = 6.2 Hz, 1H), 0.94 (d, *J* = 6.1 Hz, 1H). **^13^C NMR** (101 MHz, CDCl_3_) δ 175.4, 150.0, 132.5 (2C), 123.5 (2C), 118.9, 43.0, 37.9, 26.2, 22.7, 22.6, 17.5. **HRMS−ESI** (*m*/*z*) for C_13_H_17_BrO_2_ [M + Na]^+^: calcd 307.0304 (^79^Br), 309.0283 (^81^Br), found 307.0304 (^79^Br), 309.0288 (^81^Br).

*4-Bromophenyl 2-methylpent-4-ynoate* (**3d**): Compound **3d** was prepared according to the general procedure using **1d** (98.0 mg, 0.50 mmol, 1.00 equiv) and *p*-bromophenol (95.2 mg, 0.55 mmol, 1.10 equiv) at rt for 12 h. Purification by flash column chromatography (petroleum ether/ethyl acetate = 20/1, *v*/*v*) afforded 4-bromophenyl 2-methylpent-4-ynoate (**3d**) as a colorless oil (121.9 mg, 90%). **R*_f_*** = 0.40 (petroleum ether/ethyl acetate = 40/1, *v*/*v*). **^1^H NMR** (400 MHz, CDCl_3_) δ 7.49 (d, *J* = 8.9 Hz, 2H), 6.99 (d, *J* = 8.8 Hz, 2H), 2.91 (m, 1H), 2.59 (comp, 2H), 2.06 (s, 1H), 1.41 (d, *J* = 7.1 Hz, 3H). **^13^C NMR** (101 MHz, CDCl_3_) δ 173.1, 149.9, 132.6 (2C), 123.5 (2C), 119.1, 81.0, 70.5, 39.0, 22.9, 16.4. **HRMS−ESI** (*m*/*z*) for C_12_H_11_BrO_2_ [M + Na]^+^: calcd 288.9835 (^79^Br), 290.9815 (^81^Br), found 288.9831 (^79^Br), 290.9815 (^81^Br).

*4-Bromophenyl 3-cyclohexyl-2-methylpropanoate* (**3e**): Compound **3e** was prepared according to the general procedure using **1e** (127.0 mg, 0.50 mmol, 1.00 equiv) and *p*-bromophenol (95.2 mg, 0.55 mmol, 1.10 equiv) at rt for 12 h. Purification by flash column chromatography (petroleum ether/ethyl acetate = 20/1, *v*/*v*) afforded 4-bromophenyl 3-cyclohexyl-2-methylpropanoate (**3e**) as a colorless oil (137.7 mg, 85%). **R*_f_*** = 0.40 (petroleum ether/ethyl acetate = 40/1, *v*/*v*). **^1^H NMR** (400 MHz, CDCl_3_) δ 7.48 (d, *J* = 8.8 Hz, 2H), 6.96 (d, *J* = 8.8 Hz, 2H), 2.84–2.71 (m, 1H), 1.84–1.62 (comp, 6H), 1.37 (comp, 2H), 1.27 (d, *J* = 6.9 Hz, 3H), 1.23–1.10 (comp, 3H), 0.93 (comp, 2H). **^13^C NMR** (101 MHz, CDCl_3_) δ 175.5, 150.1, 132.6 (2C), 123.5 (2C), 118.8, 41.6, 37.2, 35.7, 33.5, 33.3, 26.7, 26.4, 26.4, 17.6. **HRMS−ESI** (*m*/*z*) for C_16_H_21_BrO_2_ [M + Na]^+^: calcd 347.0617 (^79^Br), 349.0596 (^81^Br), found 347.0617 (^79^Br), 349.0598 (^81^Br).

*4-Bromophenyl 2-methyl-3-phenylpropanoate* (**3f**): Compound **3f** was prepared according to the general procedure using **1f** (124.0 mg, 0.50 mmol, 1.00 equiv) and *p*-bromophenol (95.2 mg, 0.55 mmol, 1.10 equiv) at rt for 12 h. Purification by flash column chromatography (petroleum ether/ethyl acetate = 20/1, *v*/*v*) afforded 4-bromophenyl 2-methyl-3-phenylpropanoate (**3f**) as a colorless oil (145.2 mg, 91%). **R*_f_*** = 0.40 (petroleum ether/ethyl acetate = 40/1, *v*/*v*). **^1^H NMR** (400 MHz, CDCl_3_) δ 7.44 (d, *J* = 8.4 Hz, 2H), 7.31 (m, 2H), 7.23 (comp, 3H), 6.79 (d, *J* = 8.3 Hz, 2H), 3.09 (dd, *J* = 13.3, 7.7 Hz, 1H), 2.99 (m, 1H), 2.83 (dd, *J* = 13.3, 6.6 Hz, 1H), 1.32 (d, *J* = 6.6 Hz, 3H). **^13^C NMR** (101 MHz, CDCl_3_) δ 174.4, 149.8, 139.0, 132.5 (2C), 129.2 (2C), 128.6 (2C), 126.7, 123.4 (2C), 118.9, 41.8, 40.0, 17.1. **HRMS−ESI** (*m*/*z*) for C_16_H_15_BrO_2_ [M + Na]^+^: calcd 341.0148 (^79^Br), 343.0128 (^81^Br), found 341.0149 (^79^Br), 343.0132 (^81^Br).

*4-Bromophenyl 3-(4-fluorophenyl)-2-methylpropanoate* (**3g**): Compound **3g** was prepared according to the general procedure using **1g** (133.0 mg, 0.50 mmol, 1.00 equiv) and *p*-bromophenol (95.2 mg, 0.55 mmol, 1.10 equiv) at rt for 12 h. Purification by flash column chromatography (petroleum ether/ethyl acetate = 20/1, *v*/*v*) afforded 4-bromophenyl 3-(4-fluorophenyl)-2-methylpropanoate (**3g**) as a colorless oil (151.7 mg, 95%). **R*_f_*** = 0.40 (petroleum ether/ethyl acetate = 40/1, *v*/*v*). **^1^H NMR** (400 MHz, CDCl_3_) δ 7.46 (d, *J* = 8.8 Hz, 2H), 7.19 (dd, *J* = 8.3, 5.4 Hz, 2H), 7.00 (t, *J* = 8.5 Hz, 2H), 6.81 (d, *J* = 8.8 Hz, 2H), 3.07 (dd, *J* = 13.4, 7.7 Hz, 1H), 2.94 (m, 1H), 2.81 (dd, *J* = 13.4, 6.8 Hz, 1H), 1.31 (d, *J* = 6.9 Hz, 3H). **^19^F NMR** (377 MHz, CDCl_3_) δ –116.4. **^13^C NMR** (101 MHz, CDCl_3_) δ 174.2, 161.9 (d, *J_C−F_* = 244.7 Hz), 149.8, 134.7 (d, *J_C−F_* = 3.2 Hz), 132.6 (2C), 130.6 (d, *J_C−F_* = 7.8 Hz, 2C), 123.4 (2C), 119.0, 115.5 (d, *J_C−F_* = 21.2 Hz, 2C), 41.9, 39.1, 17.0. **HRMS−ESI** (*m*/*z*) for C_16_H_14_BrFO_2_ [M + Na]^+^: calcd 359.0053 (^79^Br), 361.0033 (^81^Br), found 359.0052 (^79^Br), 361.0035 (^81^Br).

*4-Bromophenyl 3-(3-fluorophenyl)-2-methylpropanoate* (**3h**): Compound **3h** was prepared according to the general procedure using **1h** (133.0 mg, 0.50 mmol, 1.00 equiv) and *p*-bromophenol (95.2 mg, 0.55 mmol, 1.10 equiv) at rt for 12 h. Purification by flash column chromatography (petroleum ether/ethyl acetate = 20/1, *v*/*v*) afforded 4-bromophenyl 3-(3-fluorophenyl)-2-methylpropanoate (**3h**) as a colorless oil (146.7 mg, 87%). **R*_f_*** = 0.40 (petroleum ether/ethyl acetate = 40/1, *v*/*v*). **^1^H NMR** (400 MHz, CDCl_3_) δ 7.46 (d, *J* = 8.8 Hz, 2H), 7.26 (m, 1H), 7.00 (m, 1H), 6.95 (comp, 2H), 6.83 (d, *J* = 8.4 Hz, 2H), 3.10 (dd, *J* = 13.4, 7.7 Hz, 1H), 2.98 (m, 1H), 2.82 (dd, *J* = 13.4, 6.9 Hz, 1H), 1.32 (d, *J* = 6.9 Hz, 3H). **^19^F NMR** (377 MHz, CDCl_3_) δ –113.3. **^13^C NMR** (101 MHz, CDCl_3_) δ 174.0, 162.9 (d, *J_C−F_* = 246.0 Hz), 149.6, 141.4 (d, *J_C−F_* = 7.2 Hz), 132.4 (2C), 130.0 (d, *J_C−F_* = 8.3 Hz), 124.7 (d, *J_C−F_* = 2.9 Hz), 123.2 (2C), 118.9, 115.9 (d, *J_C−F_* = 21.1 Hz), 113.6 (d, *J_C−F_* = 21.1 Hz), 41.4, 39.4, 17.0. **HRMS−ESI** (*m*/*z*) for C_16_H_14_BrFO_2_ [M + Na]^+^: calcd 359.0053 (^79^Br), 361.0033 (^81^Br), found 359.0052 (^79^Br), 361.0032 (^81^Br).

*4-Bromophenyl 3-(2fluorophenyl)-2-methylpropanoate* (**3i**): Compound **3i** was prepared according to the general procedure using **1i** (133.0 mg, 0.50 mmol, 1.00 equiv) and *p*-bromophenol (95.2 mg, 0.55 mmol, 1.10 equiv) at rt for 12 h. Purification by flash column chromatography (petroleum ether/ethyl acetate = 20/1, *v*/*v*) afforded 4-bromophenyl 3-(2-fluorophenyl)-2-methylpropanoate (**3i**) as a colorless oil (143.3 mg, 93%). **R*_f_*** = 0.40 (petroleum ether/ethyl acetate = 40/1, *v*/*v*). **^1^H NMR** (400 MHz, CDCl_3_) δ 7.45 (d, *J* = 8.8 Hz, 2H), 7.25–7.17 (comp, 2H), 7.11–7.00 (comp, 2H), 6.82 (d, *J* = 8.8 Hz, 2H), 3.14–2.99 (comp, 2H), 2.97–2.85 (m, 1H), 1.33 (d, *J* = 6.6 Hz, 3H). **^19^F NMR** (377 MHz, CDCl_3_) δ –117.6. **^13^C NMR** (101 MHz, CDCl_3_) δ 174.1, 161.4 (d, *J_C−F_* = 245.3 Hz), 149.7, 132.4 (2C), 131.5 (d, *J_C−F_* = 4.7 Hz), 128.5 (d, *J_C−F_* = 8.2 Hz), 125.8 (d, *J_C−F_* = 15.7 Hz), 124.1 (d, *J_C−F_* = 3.6 Hz), 123.3 (2C), 118.8, 115.4 (d, *J_C−F_* = 22.1 Hz), 40.2, 33.2 (d, *J_C−F_* = 2.1 Hz), 17.0. **HRMS−ESI** (*m*/*z*) for C_16_H_14_BrFO_2_ [M + Na]^+^: 359.0053 (^79^Br), 361.0033 (^81^Br), found 359.0054 (^79^Br), 361.0033 (^81^Br).

*4-Bromophenyl 3-cyclohexyl-2-methylpropanoate* (**3j**): Compound **3j** was prepared according to the general procedure using **1j** (158.0 mg, 0.50 mmol, 1.00 equiv) and *p*-bromophenol (95.2 mg, 0.55 mmol, 1.10 equiv) at rt for 12 h. Purification by flash column chromatography (petroleum ether/ethyl acetate = 20/1, *v*/*v*) afforded 4-bromophenyl 3-cyclohexyl-2-methylpropanoate (**3j**) as a colorless oil (168.0 mg, 93%). **R*_f_*** = 0.40 (petroleum ether/ethyl acetate = 40/1, *v*/*v*). **^1^H NMR** (400 MHz, CDCl_3_) δ 7.57 (d, *J* = 7.9 Hz, 2H), 7.46 (d, *J* = 8.8 Hz, 2H), 7.35 (d, *J* = 7.9 Hz, 2H), 6.80 (d, *J* = 8.9 Hz, 2H), 3.16 (dd, *J* = 13.4, 7.7 Hz, 1H), 3.01 (m, 1H), 2.88 (dd, *J* = 13.4, 7.0 Hz, 1H), 1.34 (d, *J* = 6.8 Hz, 3H). **^19^F NMR** (377 MHz, CDCl_3_) δ –62.8. **^13^C NMR** (101 MHz, CDCl_3_) δ 173.9, 149.7, 143.1, 132.6 (2C), 129.5 (2C), 129.1 (q, *J_C−F_* = 32.1 Hz), 125.6 (q, *J_C−F_* = 3.8 Hz, 2C), 124.3 (q, *J_C−F_* = 272.0 Hz), 123.3 (2C), 119.1, 41.5, 39.5, 17.1. **HRMS−ESI** (*m*/*z*) for C_17_H_14_BrF_3_O_2_ [M + Na]^+^: calcd 409.0021 (^79^Br), 411.0001 (^81^Br), found 409.0025 (^79^Br), 411.0014 (^81^Br).

*4-Nitrophenyl isobutyrate* (**3k**): Compound **3k** was prepared according to the general procedure using **1a** (86.0 mg, 0.50 mmol, 1.00 equiv) and *p*-nitrophenol (76.5 mg, 0.55 mmol, 1.10 equiv) at rt for 12 h. Purification by flash column chromatography (petroleum ether/ethyl acetate = 10/1, *v*/*v*) afforded 4-nitrophenyl isobutyrate (**3k**) as a white solid (88.8 mg, 85%). The NMR data of **3k** were in agreement with the literature data [109]. **R*_f_*** = 0.50 (petroleum ether/ethyl acetate = 10/1, *v*/*v*). **HRMS−ESI** (*m*/*z*) for C_10_H_11_NO_4_ [M + Na]^+^: calcd 232.0581, found 232.0580.

*4-(Trifluoromethyl)phenyl isobutyrate* (**3l**): Compound **3l** was prepared according to the general procedure using **1a** (86.0 mg, 0.50 mmol, 1.00 equiv) and 4-(trifluoromethyl)phenol (89.2 mg, 0.55 mmol, 1.10 equiv) at rt for 12 h. Purification by flash column chromatography (petroleum ether/ethyl acetate = 40/1, *v*/*v*) afforded 4-(trifluoromethyl)phenyl isobutyrate (**3l**) as a colorless oil (104.4 mg, 90%). **R*_f_*** = 0.40 (petroleum ether/ethyl acetate = 40/1, *v*/*v*). **^1^H NMR** (400 MHz, CDCl_3_) δ 7.64 (d, *J* = 8.6 Hz, 2H), 7.20 (d, *J* = 8.6 Hz, 2H), 2.82 (hept, *J* = 7.0 Hz, 1H), 1.33 (d, *J* = 7.0 Hz, 6H). **^19^F NMR** (377 MHz, CDCl_3_) δ –62.2. **^13^C NMR** (101 MHz, CDCl_3_) δ175.1, 153.4, 128.1 (q, *J_C−F_* = 31.2 Hz), 126.7 (q, *J_C−F_* = 3.8 Hz, 2C), 126.2 (q, *J_C−F_* = 271.3 Hz), 122.0 (2C), 34.2, 18.8 (2C). **HRMS−ESI** (*m*/*z*) for C_11_H_11_F_3_O_2_ [M + Na]^+^: calcd 255.0603, found 255.0607.

*4-(1H-Tetrazol-1-yl)phenyl isobutyrate* (**3m**): Compound **3m** was prepared according to the general procedure using **1a** (86.0 mg, 0.50 mmol, 1.00 equiv) and 4-(1H-tetrazol-1-yl)phenol (89.2 mg, 0.55 mmol, 1.10 equiv) at rt for 12 h. Purification by flash column chromatography (petroleum ether/ethyl acetate = 10/1, *v*/*v*) afforded 4-(1H-tetrazol-1-yl)phenyl isobutyrate (**3m**) as a colorless oil (98.7 mg, 85%). **R*_f_*** = 0.35 (petroleum ether/ethyl acetate = 10/1, *v*/*v*). **^1^H NMR** (400 MHz, CDCl_3_) δ 8.97 (s, 1H), 7.72 (d, *J* = 8.6 Hz, 2H), 7.33 (d, *J* = 8.6 Hz, 2H), 2.85 (hept, *J* = 7.0 Hz, 1H), 1.35 (d, *J* = 7.0 Hz, 6H). **^13^C NMR** (101 MHz, CDCl_3_) δ 175.1, 151.9, 140.6, 131.1, 123.5 (2C), 122.5 (2C), 34.2, 18.8 (2C). **HRMS−ESI** (*m*/*z*) for C_11_H_12_N_4_O_2_ [M + Na]^+^: calcd 255.0852, found 255.0854.

*4-(Dimethylamino)phenyl isobutyrate* (**3n**): Compound **3n** was prepared according to the general procedure using **1a** (86.0 mg, 0.50 mmol, 1.00 equiv) and 4-(dimethylamino)phenol (75.4 mg, 0.55 mmol, 1.10 equiv) at rt for 12 h. Purification by flash column chromatography (petroleum ether/ethyl acetate = 20/1, *v*/*v*) afforded 4-(dimethylamino)phenyl isobutyrate (**3n**) as a white solid (88.0 mg, 85%). **R*_f_*** = 0.40 (petroleum ether/ethyl acetate = 40/1, *v*/*v*). **mp** 59–60 °C (petroleum ether). **^1^H NMR** (400 MHz, CDCl_3_) δ 6.93 (d, *J* = 9.1 Hz, 2H), 6.70 (d, *J* = 9.0 Hz, 2H), 2.92 (s, 6H), 2.76 (hept, *J* = 7.0 Hz, 1H), 1.30 (d, *J* = 7.0 Hz, 6H). **^13^C NMR** (101 MHz, CDCl_3_) δ 176.2, 148.7, 141.8, 121.7 (2C), 113.2 (2C), 41.0 (2C), 34.1, 19.0 (2C). **HRMS−ESI** (*m*/*z*) for C_12_H_17_NO_2_ [M + H]^+^: calcd 208.1332, found 208.1334.

*4-Methoxyphenyl isobutyrate* (**3o**): Compound **3o** was prepared according to the general procedure using **1a** (86.0 mg, 0.50 mmol, 1.00 equiv) and 4-methoxyphenol (68.3 mg, 0.55 mmol, 1.10 equiv) at rt for 12 h. Purification by flash column chromatography (petroleum ether/ethyl acetate = 20/1, *v*/*v*) afforded 4-methoxyphenyl isobutyrate (**3o**) as a colorless oil (91.2 mg, 94%). The NMR data of **3o** were in agreement with the literature data [110]. **R*_f_*** = 0.40 (petroleum ether/ethyl acetate = 20/1, *v*/*v*). **HRMS−ESI** (*m*/*z*) for C_11_H_14_O_3_ [M + Na]^+^: calcd 217.0835, found 217.0837. 

*Methyl 4-(isobutyryloxy)benzoate* (**3p**): Compound **3p** was prepared according to the general procedure using **1a** (86.0 mg, 0.50 mmol, 1.00 equiv) and methyl 4-hydroxybenzoate (83.7 mg, 0.55 mmol, 1.10 equiv) at rt for 4 h. Purification by flash column chromatography (petroleum ether/ethyl acetate = 20/1, *v*/*v*) afforded methyl 4-(isobutyryloxy)benzoate (**3p**) as a colorless oil (100.0 mg, 90%). **R*_f_*** = 0.40 (petroleum ether/ethyl acetate = 40/1, *v*/*v*). **^1^H NMR** (400 MHz, CDCl_3_) δ 8.07 (d, *J* = 9.1 Hz, 2H), 7.15 (d, *J* = 9.1 Hz, 2H), 3.91 (s, 3H), 2.81 (hept, *J* = 7.0 Hz, 1H), 1.32 (d, *J* = 6.9 Hz, 6H). **^13^C NMR** (101 MHz, CDCl_3_) δ 175.1, 166.4, 154.6, 131.1 (2C), 127.6, 121.6 (2C), 52.2, 34.2, 18.9 (2C). **HRMS−ESI** (*m*/*z*) for C_12_H_14_O_4_ [M + Na]^+^: calcd 245.0785, found 245.0787.

*4-Formylphenyl isobutyrate* (**3q**): Compound **3q** was prepared according to the general procedure using **1a** (86.0 mg, 0.50 mmol, 1.00 equiv) and 4-hydroxybenzaldehyde (67.2 mg, 0.55 mmol, 1.10 equiv) at rt for 12 h. Purification by flash column chromatography (petroleum ether/ethyl acetate = 20/1, *v*/*v*) afforded 4-formylphenyl isobutyrate (**3q**) as a colorless oil (82.6 mg, 86%). **R*_f_*** = 0.40 (petroleum ether/ethyl acetate = 40/1, *v*/*v*). **^1^H NMR** (400 MHz, CDCl_3_) δ 9.99 (s, 1H), 7.92 (d, *J* = 8.5 Hz, 2H), 7.27 (d, *J* = 8.4 Hz, 2H), 2.83 (hept, *J* = 6.9 Hz, 1H), 1.35 (d, *J* = 6.9 Hz, 6H). **^13^C NMR** (101 MHz, CDCl_3_) δ 190.9, 174.9, 155.7, 133.9, 131.2 (2C), 122.3 (2C), 34.3, 18.8 (2C). **HRMS−ESI** (*m*/*z*) for C_11_H_12_O_3_ [M + Na]^+^: calcd 215.0679, found 215.0681.

*Naphthalen-1-yl isobutyrate* (**3r**): Compound **3r** was prepared according to the general procedure using **1a** (86.0 mg, 0.50 mmol, 1.00 equiv) and naphthalen-1-ol (79.3 mg, 0.55 mmol, 1.10 equiv) at rt for 12 h. Purification by flash column chromatography (petroleum ether/ethyl acetate = 20/1, *v*/*v*) afforded naphthalen-1-yl isobutyrate (**3r**) as a colorless oil (98.6mg, 92%). **R*_f_*** = 0.40 (petroleum ether/ethyl acetate = 40/1, *v*/*v*). **^1^H NMR** (400 MHz, CDCl_3_) δ 8.00–7.86 (comp, 2H), 7.77 (d, *J* = 8.3 Hz, 1H), 7.60–7.46 (comp, 3H), 7.28 (d, *J* = 7.4 Hz, 1H), 3.04 (hept, *J* = 7.0 Hz, 1H), 1.49 (d, *J* = 7.0 Hz, 6H). **^13^C NMR** (101 MHz, CDCl_3_) δ 175.6, 146.7, 134.7, 128.1, 127.0, 126.4, 126.4, 125.9, 125.4, 121.1, 118.0, 34.5, 19.2 (2C). **HRMS−ESI** (*m*/*z*) for C_14_H_14_O_2_ [M + Na]^+^: calcd 237.0886, found 237.0887.

*[1,1’-Biphenyl]-2-yl isobutyrate* (**3s**): Compound **3s** was prepared according to the general procedure using **1a** (86.0 mg, 0.50 mmol, 1.00 equiv) and [1,1’-biphenyl]-2-ol (93.6 mg, 0.55 mmol, 1.10 equiv) at rt for 12 h. Purification by flash column chromatography (petroleum ether/ethyl acetate = 20/1, *v*/*v*) afforded [1,1’-biphenyl]-2-yl isobutyrate (**3s**) as a colorless oil (111.7 mg, 93%). **R*_f_*** = 0.40 (petroleum ether/ethyl acetate = 40/1, *v*/*v*). **^1^H NMR** (400 MHz, CDCl_3_) δ 7.45–7.29 (comp, 8H), 7.13 (d, *J* = 7.9 Hz, 1H), 2.62 (hept, *J* = 6.9 Hz, 1H), 1.10 (d, *J* = 6.8 Hz, 6H). **^13^C NMR** (101 MHz, CDCl_3_) δ 175.4, 147.9, 137.6, 135.2, 130.9, 129.1 (2C), 128.5, 128.1 (2C), 127.4, 126.1, 122.8, 34.1, 18.7 (2C). **HRMS−ESI** (*m*/*z*) for C_16_H_16_O_2_ [M + Na]^+^: calcd 263.1042, found 263.1044.

*2,4-Dibromophenyl isobutyrate* (**3t**): Compound **3t** was prepared according to the general procedure using **1a** (86.0 mg, 0.50 mmol, 1.00 equiv) and 2,4-dibromophenol (138.5 mg, 0.55 mmol, 1.10 equiv) at rt for 12 h. Purification by flash column chromatography (petroleum ether/ethyl acetate = 20/1, *v*/*v*) afforded methyl 2,4-dibromophenyl isobutyrate (**3t**) as a colorless oil (137.2 mg, 86%). **R*_f_*** = 0.40 (petroleum ether/ethyl acetate = 40/1, *v*/*v*). **^1^H NMR** (400 MHz, CDCl_3_) δ 7.46 (d, *J* = 8.7 Hz, 1H), 7.33–7.22 (comp, 2H), 2.86 (hept, *J* = 7.0 Hz, 1H), 1.36 (d, *J* = 7.0 Hz, 6H). **^13^C NMR** (101 MHz, CDCl_3_) δ 174.2, 148.9, 134.2, 130.3, 127.1, 121.1, 115.3, 34.2, 18.9 (2C). **HRMS−ESI** (*m*/*z*) for C_10_H_10_^79^Br_2_O_2_ [M + Na]^+^: calcd 342.8940, found 342.8944; C_10_H_10_^79^Br^81^Br O_2_ [M + Na]^+^: calcd 344.8920, found 344.8919.

*Methyl 3-(isobutyryloxy)-2-naphthoate* (**3u**): Compound **3u** was prepared according to the general procedure using **1a** (86.0 mg, 0.50 mmol, 1.00 equiv) and methyl 3-hydroxy-2-naphthoate (111.2 mg, 0.55 mmol, 1.10 equiv) at rt for 12 h. Purification by flash column chromatography (petroleum ether/ethyl acetate = 20/1, *v*/*v*) afforded methyl 3-(isobutyryloxy)-2-naphthoate (**3u**) as a white solid (114.7 mg, 84%). **R*_f_*** = 0.40 (petroleum ether/ethyl acetate = 20/1, *v*/*v*). **mp** 58–59 °C (petroleum ether). **^1^H NMR** (400 MHz, CDCl_3_) δ 8.58 (s, 1H), 7.93 (d, *J* = 8.2 Hz, 1H), 7.80 (d, *J* = 8.2 Hz, 1H), 7.63–7.48 (comp, 3H), 3.92 (s, 3H), 2.93 (hept, *J* = 7.0 Hz, 1H), 1.39 (d, *J* = 7.1 Hz, 6H). **^13^C NMR** (101 MHz, CDCl_3_) δ176.1, 165.3, 147.0, 135.7, 133.7, 130.7, 129.1, 128.9, 127.3, 126.6, 122.5, 121.1, 52.3, 34.3, 19.0 (2C). **HRMS−ESI** (*m*/*z*) for C_16_H_16_O_4_ [M + Na]^+^: calcd 295.0941, found 295.0943.

*2,6-Dimethylphenyl isobutyrate* (**3v**): Compound **3v** was prepared according to the general procedure using **1a** (86.0 mg, 0.50 mmol, 1.00 equiv) and 2,6-dimethylphenol (68.2 mg, 0.55 mmol, 1.10 equiv) at 60 °C for 12 h. Purification by flash column chromatography (petroleum ether/ethyl acetate = 20/1, *v*/*v*) afforded 2,6-dimethylphenyl isobutyrate (**3v**) as a colorless oil (89.4 mg, 93%). **R*_f_*** = 0.40 (petroleum ether/ethyl acetate = 40/1, *v*/*v*). **^1^H NMR** (400 MHz, CDCl_3_) δ 7.15–6.93 (comp, 3H), 2.87 (hept, *J* = 7.0 Hz, 1H), 2.13 (s, 6H), 1.36 (d, *J* = 7.0 Hz, 6H). **^13^C NMR** (101 MHz, CDCl_3_) δ 174.7, 148.1, 130.1, 128.6 (2C), 125.7 (2C), 34.3, 19.2 (2C), 16.3 (2C). **HRMS−ESI** (*m*/*z*) for C_12_H_16_O_2_ [M + Na]^+^: calcd 215.1042, found 215.1041.

*2-Oxo-2H-chromen-7-yl isobutyrate* (**3w**): Compound **3w** was prepared according to the general procedure using **1a** (86.0 mg, 0.50 mmol, 1.00 equiv) and 7-hydroxy-2H-chromen-2-one (89.2 mg, 0.55 mmol, 1.10 equiv) at rt for 12 h. Purification by flash column chromatography (petroleum ether/ethyl acetate = 20/1, *v*/*v*) afforded 2-oxo-2H-chromen-7-yl isobutyrate (**3w**) as a white solid (98.7 mg, 85%). **R*_f_*** = 0.45 (petroleum ether/ethyl acetate = 20/1, *v*/*v*). **mp** 104–105 °C (petroleum ether). **^1^H NMR** (400 MHz, CDCl_3_) δ 7.69 (d, *J* = 9.5 Hz, 1H), 7.49 (d, *J* = 8.4 Hz, 1H), 7.10 (s, 1H), 7.04 (d, *J* = 8.4 Hz, 1H), 6.39 (d, *J* = 9.5 Hz, 1H), 2.84 (hept, *J* = 7.0 Hz, 1H), 1.34 (d, *J* = 6.9 Hz, 6H). **^13^C NMR** (101 MHz, CDCl_3_) δ 174.9, 160.4, 154.7, 153.5, 142.9, 128.5, 118.4, 116.6, 116.0, 110.4, 34.2, 18.8 (2C). **HRMS−ESI** (*m*/*z*) for C_13_H_12_O_4_ [M + Na]^+^: calcd 255.0628, found 255.0628.

*5-Bromopyridin-3-yl isobutyrate* (**3x**): Compound **3x** was prepared according to the general procedure using **1a** (86.0 mg, 0.50 mmol, 1.00 equiv) and 5-bromopyridin-3-ol (95.7 mg, 0.55 mmol, 1.10 equiv) at rt for 12 h. Purification by flash column chromatography (petroleum ether/ethyl acetate = 20/1, *v*/*v*) afforded 5-bromopyridin-3-yl isobutyrate (**3x**) as a colorless oil (100.1 mg, 82%). **R*_f_*** = 0.45 (petroleum ether/ethyl acetate = 20/1, *v*/*v*). **^1^H NMR** (400 MHz, CDCl_3_) δ 8.57 (s, 1H), 8.38 (s, 1H), 7.71 (s, 1H), 2.86 (hept, *J* = 7.0 Hz, 1H), 1.35 (d, *J* = 7.0 Hz, 6H). **^13^C NMR** (101 MHz, CDCl_3_) δ 174.6, 147.9, 147.5, 141.6, 132.1, 119.9, 34.1, 18.8 (2C). **HRMS−ESI** (*m*/*z*) for C_9_H_10_BrNO_2_ [M + H]^+^: calcd 243.9968 (^79^Br), 245.9947 (^81^Br), found 243.9970 (^79^Br), 245.9948 (^81^Br).

*2-Chloropyrimidin-5-yl isobutyrate* (**3y**): Compound **3y** was prepared according to the general procedure using **1a** (86.0 mg, 0.50 mmol, 1.00 equiv) and 2-chloropyrimidin-5-ol (71.8 mg, 0.55 mmol, 1.10 equiv) at rt for 12 h. Purification by flash column chromatography (petroleum ether/ethyl acetate = 20/1, *v*/*v*) afforded 2-chloropyrimidin-5-yl isobutyrate (**3y**) as a colorless oil (85.3 mg, 85%). **R*_f_*** = 0.45 (petroleum ether/ethyl acetate = 20/1, *v*/*v*). **^1^H NMR** (400 MHz, CDCl_3_) δ 8.50 (s, 2H), 2.87 (hept, *J* = 7.0 Hz, 1H), 1.34 (d, *J* = 7.0 Hz, 6H). **^13^C NMR** (101 MHz, CDCl_3_) δ 174.1, 157.1, 152.6 (2C), 145.1, 34.1, 18.7 (2C). **HRMS−ESI** (*m*/*z*) for C_8_H_9_ClN_2_O_2_ [M + H]^+^: calcd 201.0426, found 201.0426.

#### 3.2.3. A General Procedure for Alcohols, Thiol, Thiophenol, and Amines

In a 4.0-mL vial, 2,2,5,5-tetramethyl-1,3-dioxane-4,6-dione (**1a**) (86.0 mg, 0.50 mmol, 1.00 equiv), BTMG (94.2 mg, 0.55 mmol, 1.10 equiv) were dissolved in anhydrous NMP (625.0 μL). Then, the nucleophile (0.55 mmol, 1.10 equiv) was added, and the mixture was stirred at 60 °C for 12 h. After the reaction was completed, the mixture was diluted with EA (20 mL) and washed with 2 M aqueous HCl solution (15 mL). The organic layer was separated, washed with saturated brine (20 mL), dried over anhydrous sodium sulfate, filtered, and concentrated in vacuo. The crude product was then purified by silica gel column chromatography.

*4-Bromobenzyl isobutyrate* (**4a**): Compound **4a** was prepared according to the general procedure using **1a** (86.0 mg, 0.50 mmol, 1.00 equiv) and (4-bromophenyl)methanol (102.9 mg, 0.55 mmol, 1.10 equiv) at 60 °C for 12 h. Purification by flash column chromatography (petroleum ether/ethyl acetate = 20/1, *v*/*v*) afforded 4-bromobenzyl isobutyrate (**4a**) as a colorless oil (117.0 mg, 91%). **R*_f_*** = 0.55 (petroleum ether/ethyl acetate = 20/1, *v*/*v*). **^1^H NMR** (400 MHz, CDCl_3_) δ 7.49 (d, *J* = 8.3 Hz, 2H), 7.22 (d, *J* = 8.3 Hz, 2H), 5.06 (s, 2H), 2.59 (hept, *J* = 7.0 Hz, 1H), 1.18 (d, *J* = 7.1 Hz, 6H). **^13^C NMR** (101 MHz, CDCl_3_) δ 176.8, 135.3, 131.7 (2C), 129.7 (2C), 122.1, 65.2, 34.0, 19.0 (2C). **HRMS−ESI** (*m*/*z*) for C_11_H_13_BrO_2_ [M + Na]^+^: calcd 278.9992 (^79^Br), 280.9971 (^81^Br), found 278.9991 (^79^Br), 280.9977 (^81^Br).

*3-Phenylpropyl isobutyrate* (**4b**): Compound **4b** was prepared according to the general procedure using **1a** (86.0 mg, 0.50 mmol, 1.00 equiv) and 3-phenylpropanol (74.9 mg, 0.55 mmol, 1.10 equiv) at 60 °C for 12 h. Purification by flash column chromatography (petroleum ether/ethyl acetate = 20/1, *v*/*v*) afforded 3-phenylpropyl isobutyrate (**4b**) as a colorless oil (96.0 mg, 93%). **R*_f_*** = 0.55 (petroleum ether/ethyl acetate = 20/1, *v*/*v*). **^1^H NMR** (400 MHz, CDCl_3_) δ 7.33–7.24 (m, 2H), 7.23–7.15 (comp, 3H), 4.09 (t, *J* = 6.5 Hz, 2H), 2.69 (t, *J* = 7.7 Hz, 2H), 2.55 (hept, *J* = 7.0 Hz, 1H), 1.96 (tt, *J* = 14.0, 6.5 Hz, 2H), 1.18 (d, *J* = 7.0 Hz, 6H). **^13^C NMR** (101 MHz, CDCl_3_) δ 177.2, 141.3, 128.45 (2C), 128.41 (2C), 126.0, 63.5, 34.1, 32.2, 30.3, 19.0 (2C). **HRMS−ESI** (*m*/*z*) for C_13_H_18_O_2_ [M + Na]^+^: calcd 229.1199, found 229.1201.

*Benzhydryl isobutyrate* (**4c**): Compound **4c** was prepared according to the general procedure using **1a** (86.0 mg, 0.50 mmol, 1.00 equiv) and diphenylmethanol (101.3 mg, 0.55 mmol, 1.10 equiv) at 60 °C for 12 h. Purification by flash column chromatography (petroleum ether/ethyl acetate = 20/1, *v*/*v*) afforded benzhydryl isobutyrate (**4c**) as a colorless oil (114.4 mg, 90%). **R*_f_*** = 0.55 (petroleum ether/ethyl acetate = 20/1, *v*/*v*). **^1^H NMR** (400 MHz, CDCl_3_) δ 7.39–7.27 (comp, 10H), 6.86 (s, 1H), 2.67 (hept, *J* = 7.0 Hz, 1H), 1.21 (d, *J* = 6.9 Hz, 6H). **^13^C NMR** (101 MHz, CDCl_3_) δ 176.0, 140.5 (2C), 128.5 (4C), 127.8 (2C), 127.0 (4C), 76.6, 34.2, 18.9 (2C). **HRMS−ESI** (*m*/*z*) for C_17_H_18_O_2_ [M + Na]^+^: calcd 277.1199, found 277.1198.

*Cyclohexyl isobutyrate* (**4d**): Compound **4d** was prepared according to the general procedure using **1a** (86.0 mg, 0.50 mmol, 1.00 equiv) and cyclohexanol (55.1 mg, 0.55 mmol, 1.10 equiv) at 80 °C for 12 h. Purification by flash column chromatography (petroleum ether/ethyl acetate = 20/1, *v*/*v*) afforded cyclohexyl isobutyrate (**4d**) as a colorless oil (69.8 mg, 82%). **R*_f_*** = 0.55 (petroleum ether/ethyl acetate = 20/1, *v*/*v*). The NMR data of **4d** were in agreement with the literature data [111]. **HRMS−ESI** (*m*/*z*) for C_10_H_18_O_2_ [M + Na]^+^: calcd 193.1199, found 193.1205. 

*S-(4-Fluorophenyl) 2-methylpropanethioate* (**4e**): Compound **4e** was prepared according to the general procedure using **1a** (86.0 mg, 0.50 mmol, 1.00 equiv) and 4-fluorothiophenol (74.5 mg, 0.55 mmol, 1.10 equiv) at rt for 12 h. Purification by flash column chromatography (petroleum ether/ethyl acetate = 10/1, *v*/*v*) afforded S-(4-fluorophenyl) 2-methylpropanethioate (**4e***)* as a colorless oil (81.2 mg, 82%). **R*_f_*** = 0.65 (petroleum ether/ethyl acetate = 10/1, *v*/*v*). **^1^H NMR** (400 MHz, CDCl_3_) δ 7.41–7.33 (m, 2H), 7.14–7.05 (m, 2H), 2.86 (hept, *J* = 6.9 Hz, 1H), 1.26 (d, *J* = 6.8 Hz, 6H). **^19^F NMR** (377 MHz, CDCl_3_) δ –111.6. **^13^C NMR** (101 MHz, CDCl_3_) δ 202.0, 163.5 (d, *J* = 249.6 Hz), 136.8 (d, *J* = 8.6 Hz, 2C), 123.3, 116.5 (d, *J* = 22.2 Hz, 2C), 43.1, 19.5 (2C). **HRMS−ESI** (*m*/*z*) for C_10_H_11_FOS [M + H]^+^: calcd 199.0587, found 199.0584. 

*S-Dodecyl 2-methylpropanethioate* (**4f**): Compound **4f** was prepared according to the general procedure using **1a** (86.0 mg, 0.50 mmol, 1.00 equiv) and dodecane-1-thiol (111.3 mg, 0.55 mmol, 1.10 equiv) at rt for 12 h. Purification by flash column chromatography (petroleum ether/ethyl acetate = 3/1, *v*/*v*) afforded S-dodecyl 2-methylpropanethioate (**4f**) as a colorless oil (51.1 mg, 90%). **R*_f_*** = 0.60 (petroleum ether/ethyl acetate = 10/1, *v*/*v*). **^1^H NMR** (400 MHz, CDCl_3_) δ 2.84 (t, *J* = 7.4 Hz, 2H), 2.72 (hept, *J* = 6.9 Hz, 1H), 1.65–1.49 (m, 2H), 1.42–1.21 (comp, 18H), 1.18 (d, *J* = 7.0 Hz, 6H), 0.88 (t, *J* = 6.7 Hz, 3H). **^13^C NMR** (101 MHz, CDCl_3_) δ 204.5, 43.3, 32.1, 29.8 (3C), 29.7, 29.6, 29.5, 29.3, 29.0, 28.7, 22.8, 19.6 (2C), 14.2. **HRMS−ESI** (*m*/*z*) for C_16_H_32_OS [M + H]^+^: calcd 273.2247, found 273.2247.

*N-(4-Bromobenzyl)isobutyramide* (**4g**): Compound **4g** was prepared according to the general procedure using **1a** (86.0 mg, 0.50 mmol, 1.00 equiv) and (4-bromophenyl)methanamine (102.3 mg, 0.55 mmol, 1.10 equiv) at 100 °C for 12 h. Purification by flash column chromatography (petroleum ether/ethyl acetate = 10/1, *v*/*v*) afforded *N*-(4-bromobenzyl)isobutyramide (**4g**) as a yellow solid (117.4 mg, 92%). **R*_f_*** = 0.30 (petroleum ether/ethyl acetate = 10/1, *v*/*v*). **mp** 114–115 °C (petroleum ether). **^1^H NMR** (400 MHz, CDCl_3_) δ 7.44 (d, *J* = 8.4 Hz, 2H), 7.13 (d, *J* = 8.4 Hz, 2H), 5.83 (s, 1H), 4.37 (d, *J* = 5.9 Hz, 2H), 2.38 (hept, *J* = 6.9 Hz, 1H), 1.17 (d, *J* = 6.9 Hz, 6H). **^13^C NMR** (101 MHz, CDCl_3_) δ 177.0, 137.8, 131.9 (2C), 129.5 (2C), 121.4, 42.9, 35.8, 19.7 (2C). **HRMS−ESI** (*m*/*z*) for C_11_H_14_BrNO [M + H]^+^: calcd 256.0332 (^79^Br), 258.0312 (^81^Br), found 256.0032 (^79^Br), 258.0312 (^81^Br).

*N-(3-Phenylpropyl)isobutyramide* (**4h**): Compound **4h** was prepared according to the general procedure using **1a** (86.0 mg, 0.50 mmol, 1.00 equiv) and 3-phenylpropan-1-amine (74.4 mg, 0.55 mmol, 1.10 equiv) at 100 °C for 12 h. Purification by flash column chromatography (petroleum ether/ethyl acetate = 20/1, *v*/*v*) afforded *N*-(3-phenylpropyl)isobutyramide (**4h**) as a yellow oil (93.4 mg, 91%). **R*_f_*** = 0.20 (petroleum ether/ethyl acetate = 20/1, *v*/*v*). **^1^H NMR** (400 MHz, CDCl_3_) δ 7.39–7.24 (m, 2H), 7.23–7.12 (comp, 3H), 5.43 (br, 1H), 3.29 (q, *J* = 6.5 Hz, 2H), 2.65 (t, *J* = 7.6 Hz, 2H), 2.28 (hept, *J* = 6.9 Hz, 1H), 1.91–1.77 (m, 2H), 1.12 (d, *J* = 6.9 Hz, 6H). **^13^C NMR** (101 MHz, CDCl_3_) δ 177.0, 141.7, 128.6 (2C), 128.5 (2C), 126.2, 39.2, 35.8, 33.5, 31.4, 19.7 (2C). **HRMS−ESI** (*m*/*z*) for C_13_H_19_NO [M + H]^+^: calcd 206.1540, found 206.1540. 

*N-(Heptadecan-9-yl)isobutyramide* (**4i**): Compound **4i** was prepared according to the general procedure using **1a** (86.0 mg, 0.50 mmol, 1.00 equiv) and heptadecan-9-amine (140.5 mg, 0.55 mmol, 1.10 equiv) at 100 °C for 12 h. Purification by flash column chromatography (petroleum ether/ethyl acetate = 20/1, *v*/*v*) afforded *N-(heptadecan-9-yl)isobutyramide* (**4i**) as a white solid (146.5 mg, 90%). **R*_f_*** = 0.50 (petroleum ether/ethyl acetate = 10/1, *v*/*v*). **mp** 85–86 °C (petroleum ether). **^1^H NMR** (400 MHz, CDCl_3_) δ 5.05 (d, *J* = 9.2 Hz, 1H), 3.88 (m, 1H), 2.31 (hept, *J* = 6.9 Hz, 1H), 1.47 (m, 2H), 1.27 (comp, 26H), 1.15 (d, *J* = 6.9 Hz, 6H), 0.87 (t, *J* = 6.7 Hz, 6H). **^13^C NMR** (101 MHz, CDCl_3_) δ 176.5, 49.0, 36.1, 35.5 (2C), 32.0 (2C), 29.75 (2C), 29.67 (2C), 29.4 (2C), 26.0 (2C), 22.8 (2C), 19.9 (2C), 14.2 (2C). **HRMS−ESI** (*m*/*z*) for C_21_H_43_NO [M + H]^+^: calcd 326.3418, found 326.3418. 

*N-Cyclohexylisobutyramide* (**4j**): Compound **4j** was prepared according to the general procedure using **1a** (86.0 mg, 0.50 mmol, 1.00 equiv) and cyclohexanamine (54.5 mg, 0.55 mmol, 1.10 equiv) at 100 °C for 12 h. Purification by flash column chromatography (petroleum ether/ethyl acetate = 10/1, *v*/*v*) afforded *N*-cyclohexylisobutyramide (**4j**) as a white solid (71.9 mg, 85%). **R*_f_*** = 0.50 (petroleum ether/ethyl acetate = 10/1, *v*/*v*). **mp** 116–117 °C (petroleum ether). **^1^H NMR** (400 MHz, CDCl_3_) δ 5.36 (s, 1H), 3.79–3.64 (m, 1H), 2.28 (hept, *J* = 6.8 Hz, 1H), 1.88 (m, 2H), 1.75–1.64 (m, 2H), 1.64–1.50 (m, 1H), 1.36 (m, 2H), 1.12 (comp, 9H). **^13^C NMR** (101 MHz, CDCl_3_) δ 176.1, 47.9, 35.9, 33.3 (2C), 25.7, 25.0 (2C), 19.8 (2C). **HRMS−ESI** (*m*/*z*) for C_10_H_19_NO [M + H]^+^: calcd 170.1540, found 170.1540.

*2-Methyl-1-morpholinopropan-1-one* (**4k**): Compound **4k** was prepared according to the general procedure using **1a** (86.0 mg, 0.50 mmol, 1.00 equiv) and morpholine (47.9 mg, 0.55 mmol, 1.10 equiv) at 100 °C for 12 h. Purification by flash column chromatography (petroleum ether/ethyl acetate = 3/1, *v*/*v*) afforded 2-methyl-1-morpholinopropan-1-one (**4k**) as a colorless oil (51.1 mg, 62%). **R*_f_*** = 0.45 (petroleum ether/ethyl acetate = 3/1, *v*/*v*). **^1^H NMR** (400 MHz, CDCl_3_) δ 3.74–3.63 (comp, 4H), 3.63–3.56 (m, 2H), 3.50 (m, 2H), 2.74 (hept, *J* = 6.8 Hz, 1H), 1.11 (d, *J* = 6.8 Hz, 6H). **^13^C NMR** (101 MHz, CDCl_3_) δ 175.7, 67.1, 66.9, 46.1, 42.2, 30.0, 19.4 (2C). **HRMS−ESI** (*m*/*z*) for C_8_H_15_NO_2_ [M + H]^+^: calcd 158.1176, found 158.1176.

#### 3.2.4. Gram-Scale Preparation of 4-Acetamidophenyl Isobutyrate (**3z**)

To a 50-mL round-bottom flask, 5,5-dimethyl-Meldrum’s acid **1a** (1.72 g, 10.0 mmol, 1.00 equiv), BTMG (34.2 mg, 0.20 mmol, 0.02 equiv) were dissolved in anhydrous NMP (25.0 mL). Then, acetaminophen (1.66 g, 11.0 mmol, 1.10 equiv) was added and stirred at 60 °C for 12 h. After the reaction was completed, the mixture was diluted with EA (100 mL) and washed with 2 × 2 M aqueous HCl solution (50 mL). The organic layer was separated, washed with saturated brine (100 mL), dried over anhydrous sodium sulfate, filtered, and concentrated in vacuo. The crude product was then purified by silica gel column chromatography (petroleum ether/ethyl acetate = 3/1, *v*/*v*), which afforded 4-acetamidophenyl isobutyrate (**3z**) as a white solid (2.12 g, 96%). **R*_f_*** = 0.35 (petroleum ether/ethyl acetate = 3/1, *v*/*v*). **mp** 111−112 °C (petroleum ether). **^1^H NMR** (400 MHz, CDCl_3_) δ 7.51 (d, *J* = 8.4 Hz, 2H), 7.23 (s, 1H), 7.04 (d, *J* = 8.4 Hz, 2H), 2.81 (hept, *J* = 7.0 Hz, 1H), 2.19 (s, 3H), 1.33 (d, *J* = 7.0 Hz, 6H). **^13^C NMR** (101 MHz, CDCl_3_) δ 176.1, 168.5, 147.0, 135.6, 121.8 (2C), 120.9 (2C), 34.1, 24.4, 18.9 (2C). **HRMS−ESI** (*m*/*z*) for C_12_H_15_NO_3_ [M + H]^+^: calcd 222.1125, found 222.1126.

#### 3.2.5. Synthesis of 4-Acetamidophenyl 2-(((benzyloxy)carbonyl)amino)-2-Methylpropanoate (**6**)

Step 1: To a 25-mL round-bottom flask, 2,2,5,5-tetramethyl-1,3-dioxane-4,6-dione (**1a**) (688.3 mg, 4.0 mmol, 1.00 equiv) and BTMG (753.6 mg, 0.55 mmol, 1.10 equiv) were dissolved in anhydrous DCM (10.0 mL). Then, acetaminophen (665.1 mg, 4.4 mmol, 1.10 equiv) was added, and the mixture was stirred at rt for 4 h. After the reaction was completed, the mixture was diluted with DCM (20 mL) and washed with 2 M aqueous HCl solution (15 mL). The organic layer was separated, washed with saturated brine (20 mL), dried over anhydrous sodium sulfate, filtered, and concentrated in vacuo. The crude product was then purified by silica gel column chromatography (ethyl acetate), which afforded 3-(4-acetamidophenoxy)-2,2-dimethyl-3-oxopropanoic acid (**5**) as a white solid (912.5 mg, 86%). **R*_f_*** = 0.20 (dichloromethane /methanol = 40/1, *v*/*v*). **mp** 166–168 °C (petroleum ether). **^1^H NMR** (400 MHz, DMSO-*d*_6_) δ 13.04 (br, s, 1H), 9.98 (s, 1H), 7.61 (d, *J* = 9.0 Hz, 2H), 7.00 (d, *J* = 8.9 Hz, 2H), 2.04 (s, 3H), 1.46 (s, 6H). **^13^C NMR** (101 MHz, DMSO-*d_6_*) δ 173.4, 171.6, 168.2, 145.5, 137.1, 121.4 (2C), 119.9 (2C), 49.4, 23.9, 22.4 (2C). **HRMS−ESI** (*m*/*z*) for C_13_H_16_NO_5_ [M + H]^+^: calcd 266.1023, found 266.1030.

Step 2: To a 4.0-mL vial, the malonate half-ester **5** (132.5 mg, 0.50 mmol, 1.00 equiv), DPPA (165.1 mg, 0.60 mmol, 1.20 equiv), and TEA (75.9 mg, 0.75 mmol, 1.50 equiv) were dissolved in anhydrous toluene (2.5 mL). The mixture was stirred at 100 °C for 12 h. After it was cooled to rt, benzyl alcohol (108.1 mg, 1.00 mmol, 2.00 equiv) was added and stirred at 100 °C for 12 h. After the reaction was completed, the mixture was diluted with EA (20 mL) and washed with 2 M aqueous HCl solution (15 mL). The organic layer was separated, washed with saturated brine (20 mL), dried over anhydrous sodium sulfate, filtered, and concentrated in vacuo. The crude product was then purified by silica gel column chromatography (petroleum ether/ethyl acetate = 1/1, *v*/*v*), which afforded 4-acetamidophenyl 2-(((benzyloxy)carbonyl)amino)-2-methylpropanoate (**6**) as a white solid (140.6 mg, 76%). **R*_f_*** = 0.35 (petroleum ether/ethyl acetate = 1/1, *v*/*v*). **mp** 189–190 °C (petroleum ether). **^1^H NMR** (400 MHz, DMSO-*d_6_*) δ 9.97 (s, 1H), 7.98 (s, 1H), 7.57 (d, *J* = 8.4 Hz, 2H), 7.41–7.26 (comp, 5H), 6.90 (d, *J* = 8.5 Hz, 2H), 5.08 (s, 2H), 2.05 (s, 3H), 1.49 (s, 6H). **^13^C NMR** (101 MHz, DMSO-*d_6_*) δ 173.3, 168.2, 155.3, 145.9, 136.9, 136.8, 128.3 (2C), 127.8 (3C), 121.6 (2C), 119.8 (2C), 65.4, 55.6, 24.9 (2C), 23.9. **HRMS−ESI** (*m*/*z*) for C_20_H_22_N_2_O_5_ [M + H]^+^: calcd 371.1602, found 371.1610.

#### 3.2.6. One-Pot Preparation of 4-Acetamidophenyl 3-((4-bromobenzyl)amino)-2,2-dimethyl-3-oxopropanoate (**7**)

In a 4.0-mL vial, 2,2,5,5-tetramethyl-1,3-dioxane-4,6-dione (1a) (86.0 mg, 0.50 mmol, 1.00 equiv) and BTMG (94.2 mg, 0.55 mmol, 1.10 equiv) were dissolved in anhydrous DCM (1.25 mL). Then, acetaminophen (83.1 mg, 0.55 mmol, 1.10 equiv) was added, and the mixture was stirred at rt for 4 h. Then, *p*-BrBnNH_2_ (102.3 mg, 0.55 mmol, 1.10 equiv), EDC·HCl (143.8 mg, 0.75 mmol, 1.50 equiv), and HOBt (101.3 mg, 0.75 mmol, 1.50 equiv) were added, and the mixture was stirred at rt for 6 h. After the reaction was completed, the mixture was diluted with EA (20 mL) and washed with 2 M aqueous HCl solution (15 mL). The organic layer was separated, washed with saturated brine (20 mL), dried over anhydrous sodium sulfate, filtered, and concentrated in vacuo. The crude product was then purified by silica gel column chromatography (petroleum ether/ethyl acetate = 3/1, *v*/*v*), which afforded 4-acetamidophenyl 3-((4-bromobenzyl)amino)-2,2-dimethyl-3-oxopropanoate (**7**) as a white solid (140.4 mg, 65%). **R*_f_*** = 0.35 (petroleum ether/ethyl acetate = 3/1, *v*/*v*). **mp** 179–180 °C (petroleum ether). **^1^H NMR** (400 MHz, DMSO-*d_6_*) δ 9.97 (s, 1H), 8.50 (t, *J* = 6.0 Hz, 1H), 7.59 (d, *J* = 8.9 Hz, 2H), 7.48 (d, *J* = 8.3 Hz, 2H), 7.23 (d, *J* = 8.4 Hz, 2H), 6.95 (d, *J* = 8.9 Hz, 2H), 4.29 (d, *J* = 5.9 Hz, 1H), 2.04 (s, 3H), 1.49 (s, 6H). **^13^C NMR** (101 MHz, DMSO-*d_6_*) δ 172.2, 171.5, 168.2, 145.6, 139.0, 137.0, 131.0 (2C), 129.2 (2C), 121.5 (2C), 119.8 (2C), 119.7, 50.0, 41.9, 23.9, 22.7 (2C). **HRMS−ESI** (*m*/*z*) for C_20_H_21_BrN_2_O_4_ [M + H]^+^: calcd 433.0758 (^79^Br), 435.0737 (^81^Br), found 433.0766 (^79^Br), 435.0745 (^81^Br).

## 4. Conclusions

We have identified disubstituted Meldrum’s acid as a novel carbon-based scaffold with SuFEx-like reactivity. As exemplified in other SuFEx reactions, phenols are the optimal nucleophilic exchange partners. Notably, thiols and thiophenols, typically prone to oxidation by S(VI) electrophiles in classical SuFEx reactions, exhibit comparable reactivity to phenols in our method. In contrast, alcohols and amines require elevated temperatures to achieve full conversion. Sterically hindered nucleophiles, such as tertiary alcohols or bulky primary and secondary amines, remain challenging and will require the development of new catalytic systems. Alternative activation strategies for disubstituted Meldrum’s acids are currently under investigation in our laboratory and will be reported in due course.

## 5. Patents

Sun Yat-sen University has filed a patent application.

## Data Availability

Data is contained within the article or Appendix A.

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
