# Peer review of "Disubstituted Meldrum’s Acid: Another Scaffold with SuFEx-like Reactivity"

_molecules, 2025, doi:10.3390/molecules30173534_

Round 1

Reviewer 1 Report

Comments and Suggestions for Authors

Chen and co-workers report a practical esterification and amidation method based on dialkyl Meldrum’s acids. This protocol does not require conventional coupling reagents and tolerates a broad range of solvents, including water, which is particularly important for the synthesis of bioactive molecules. Moreover, the observed reactivity trend among different nucleophiles is very interesting: phenols react the fastest, whereas alcohols and amines are relatively slower and require heating. Notably, similar trends have been seen in many SuFEx reactions. Remarkably, the method is also compatible with thiols and thiophenols, which were problematic in many SuFEx reactions. I believe that this methodology offers a new approach for coupling carboxyl acid fragment with diverse nucleophiles and extending traditional SuFEx chemistry to a carbon-centered scaffold.
The authors propose a nucleophilic substitution–decarboxylation mechanism, which is well-supported through stepwise experiments. The fact that the reaction proceeds at relatively low temperatures further excludes the possibility of ketene formation under thermal conditions. In addition, the demonstration of a catalytic version of the reaction opens up opportunities for large-scale synthesis and even asymmetric catalysis. Overall, this work significantly expands the conceptual scope of SuFEx chemistry and provides a complementary and reliable coupling method for ester and amide synthesis. I recommend the manuscript be published in Molecules after minor revision.

Whether the authors explored the use of a chiral catalyst in this reaction. If so, what outcome was obtained?

Author Response

Comment 1: Whether the authors explored the use of a chiral catalyst in this reaction. If so, what outcome was obtained?

Response: Thanks for your suggestion. Indeed, we have conducted several experiments with various chiral catalysts available in my laboratroy. Unfortunately, we have not obtained promising results so far.

Reviewer 2 Report

Comments and Suggestions for Authors

This work presents a methodology that encompasses phenolic addition to and subsequent ring-opening of dialkylated Meldrum's acid followed by decarboxylation to form a substrate pool of mostly esters and amides. Overall, the methodology is exhaustive enough to demonstrate substrate tolerance (both with phenols/alcohols/amines/thiols and alkylated MA derivatives), characterization is complete with proof of purity and structure and is supported with catalytic scalability. While the authors have provided an exhaustive background on SuFEx chemistry, the comparison to this methodology might be a bit of a stretch (albeit the authors' mention this as well). The work presented is simply a preparation of functionalized esters with dialkylated MA derivatives. One can argue the substrates pool of esters prepared can mostly be prepared with transesterification or simply nucleophilic additions with acyl chlorides. However, there is an inherent advantage to the prepared work in comparison to these approaches for a number of reasons (more stability, more diverse substrates, etc.). It is the reviewer's recommendation that this work should be published by toning down the use of SuFEx-like reactivity and focus the efforts to a new esterification methodology. Some minor points to also address are detailed below that should improve the manuscript. 

  1. Scheme 2 is not a 'scheme' per se. The table/figure is a bit confusing as well as there are two sets of rows for stability but only one column for 'stability'. The table/figure just needs to be refined.
  2. In all the Tables/Schemes, the alphanumeric compound designations are not in boldface font.
  3. Table 1 heading indicates a 'Staudinger Reduction of Methyl 4-Azidobenzoate' which is not correctly depicted in the chemical reaction scheme. 
  4. Scheme 3: the random use of mmol, grams, equiv is confusing and could be cleaned up to better represent their point of scalability and catalytic activity. 

Author Response

Responses to Reviewer #2:

Reviewer 2:

Comment 1: While the authors have provided an exhaustive background on SuFEx chemistry, the comparison to this methodology might be a bit of a stretch (albeit the authors' mention this as well). The work presented is simply a preparation of functionalized esters with dialkylated MA derivatives. One can argue the substrates pool of esters prepared can mostly be prepared with transesterification or simply nucleophilic additions with acyl chlorides. However, there is an inherent advantage to the prepared work in comparison to these approaches for a number of reasons (more stability, more diverse substrates, etc.). It is the reviewer's recommendation that this work should be published by toning down the use of SuFEx-like reactivity and focus the efforts to a new esterification methodology.

Response: Thank you for your comment. As you mentioned, we have already noted “Although Meldrum’s acid may seem unrelated to SuFEx-like reactivity at first glance,” in the manuscript. However, what we really want to emphasize is the inherent similarity between disubstituted Meldrum’s acid and many SuFEx buiding blocks. This similarity (both are highly eletrophilic and sterically hindered around the central atom) leads to a similar reactivity order in nucleophilic exchange reactions. This reactivity trend has not been observed in the chemistry of acid chlorides and sulfonyl chlorides.

Comment 2: Scheme 2 is not a 'scheme' per se. The table/figure is a bit confusing as well as there are two sets of rows for stability but only one column for 'stability'. The table/figure just needs to be refined.

Response: Thank you for your comment. We apologized for the confusion. Scheme 2 have been revised and renamed in the manuscript.

Comment 3: In all the Tables/Schemes, the alphanumeric compound designations are not in boldface font.

Response: Thank you for your comment. We have revised the corresponding contents in the manuscript.

Comment 4: Table 1 heading indicates a 'Staudinger Reduction of Methyl 4-Azidobenzoate' which is not correctly depicted in the chemical reaction scheme.

Response: Thank you for your comment. We apologized for the mistake. The correct title of Table 1 has been placed in the manuscript.

Comment 5: Scheme 3: the random use of mmol, grams, equiv is confusing and could be cleaned up to better represent their point of scalability and catalytic activity.

Response: Thanks for your suggestion. We have revised Scheme 3 to make it clearer.

Reviewer 3 Report

Comments and Suggestions for Authors

The submitted manuscript Molecules-3831127, titled “Disubstituted Meldrum's acid: Another scaffold with SuFEx-like reactivity” presents a concise and elegant method for the use of disubstituted Meldrum's acid as a carbon scaffold with SuFEx-like reactivity, enabling nucleophilic (O, S and N) exchange transformations under mild conditions (BTMG/DBU), with efficient decarboxylation and good hydrolytic stability.

The introduction is well-written, and provides a solid background on the use of Meldrum’s acid and multicomponent reactions, while clearly highlighting the synthetic value and potential of the target heterocycles. The methodology is synthetically appealing and operationally simple, and the scope demonstrated in the manuscript shows promising diversity.

That said, in order to strengthen the claim of generality and versatility of the methodology, I believe the manuscript requires a revision before it can be considered for publication. The key issue relates to the structural diversity of the Meldrum’s acid derivatives employed:

1) I request the incorporation of at least one of the following substrates into the reaction scope: 5-phenyl-Meldrum's acid and/or 5-phenyl-5-methyl-Meldrum's acid.

Author Response

Responses to Reviewer #3:

Reviewer 3:

Comment 1: I request the incorporation of at least one of the following substrates into the reaction scope: 5-phenyl-Meldrum's acid and/or 5-phenyl-5-methyl-Meldrum's acid.

Response: Thanks for your suggestion. During our study, we have prepared 5-phenyl-Meldrum's acid and 5-phenyl-5-methyl-Meldrum's acid. In the case of 5-phenyl-Meldrum's acid, the reaction was not as clean as disubstituted Meldrum’s acid. Indeed, the reactions with other mono-substituted Meldrum’s acids had similar outcome since 5-H can be deprotonated under basic condition. This is also the one of the reasons why we chose disubstituted Meldrum’s acids as target substrates. For 5-phenyl-5-methyl-Meldrum's acid, we obtained 86% isolated yield under our standard conditions. At this time, we would like to keep the orignial substrate scope for disubstituted Meldrum’s acid (dialkyl ones only) since the preparation of other aryl-substituted Meldrum’s acids is not straight-forward compared to alkyl-substituted ones. However, we will continue to explore the substrates such as 5-aryl-5-alkyl and even 5,5’-diaryl Meldrum’s acids. Currently, we are working on a project of the direct arylation of Meldrum’s acids. Hopefully, we will have more results to report as soon as these substrates can be readily accessed.

Reviewer 4 Report

Comments and Suggestions for Authors

The manuscript presents an interesting concept of using Meldrum acid derivatives in mild coupling reaction with O, N and S nucleophiles in the manner resembling SuFEx chemistry. Although the conditions required for amines are not so mild (100 C), the method generally gives excellent yields and it may be useful in biological studies, as well as a general mild method for esterification and amidation. The concept of research has been properly presented and justified and the results supported with a wide range  of examples. The obtained compounds have been properly characterized. I recommend publication of this manuscript after minor revisions:

1) Scheme 2: In the left column, the word “Stability” should be below “Electrophilicity”, not next to it.

2) Table 1: In entries 4, 5 and 11 MeCN was used as solvent, so footnote “d” in these cases is probably a mistake; footnote “e” (entry 11) has not been defined.

3) The general conclusion from the different reaction conditions required for different nucleophile classes seems to be that reactivity is determined mainly by acidity of a given nucleophile rather than nucleophilicity of its heteroatom (phenols and thiols are more acidic than alcohols, and alcohols more than amines, although nitrogen is generally more nucleophilic than oxygen). Authors should comment on this observation, also in the manuscript text.

4) The possibility of the catalytic process (2 mol%) is very attractive, but it needs to be more investigated. What about the yield of a catalytic reaction at rt? Why exactly 2 mol% of base, has it been optimized?   

5) Please capitalize the names of compounds in the experimental section.

Author Response

Responses to Reviewer #4:

Reviewer 4:

Comment 1: Scheme 2: In the left column, the word “Stability” should be below “Electrophilicity”, not next to it.

Response: Thanks for your comment. We have revised Table 1 (the previous Scheme 2) in the manuscript.

Comment 2: Table 1: In entries 4, 5 and 11 MeCN was used as solvent, so footnote “d” in these cases is probably a mistake; footnote “e” (entry 11) has not been defined.

Response: Thanks for your comment. We apologized for the mistakes. We have checked and revised the correponding footnotes in the manuscript.

Comment 3: The general conclusion from the different reaction conditions required for different nucleophile classes seems to be that reactivity is determined mainly by acidity of a given nucleophile rather than nucleophilicity of its heteroatom (phenols and thiols are more acidic than alcohols, and alcohols more than amines, although nitrogen is generally more nucleophilic than oxygen). Authors should comment on this observation, also in the manuscript text.

Response: Thanks for your comment. We agree that the acidity of a nucleophile is related to the reactivity, but the acidity is related to the nucleophilicity as well. Theorectically, the nucleophility of a specific nucleophile is related to its acidity and its bulkiness. The bulkier the corresponding electrophile is, the more the bulkiness of a nucleophile affects the nucleophilicity. Therefore, the nucleophilicity of a nucleophile is related to the encountering electrophile while the acidity is not. For example, the nucleophilic exchange reactions of S(VI)-F is more Nu-sensitive than the ones of acyl fluorides. In our case, 2,6-dimethylphenol, whose pKa is similar to phenol, requires higher temperature to complete the reaction. We think that the nucleophilicity instead of the acidity will give the readers a more comprehensive picture about this reaction.

Comment 4: The possibility of the catalytic process (2 mol%) is very attractive, but it needs to be more investigated. What about the yield of a catalytic reaction at rt? Why exactly 2 mol% of base, has it been optimized?

Response: Thanks for your comment. We have conducted a brief optimization for this reaction. Indeed, catalytic reactions proceeded slower at room temperature. The condition with 2 mol% catalyst gives a good balance between catalyst loading and rates.

Comment 5: Please capitalize the names of compounds in the experimental section.

Response: Thanks for your suggestion. We have captitalized the names in the manuscript.